# Longitudinal piezoelectric resonant photoelastic modulator for efficient intensity modulation at megahertz frequencies

Okan Atalar [1✉], Raphaël Van Laer[2,3], Amir H. Safavi-Naeini [2] & Amin Arbabian [1]

Intensity modulators are an essential component in optics for controlling free-space beams. Many applications require the intensity of a free-space beam to be modulated at a single frequency, including wide-field lock-in detection for sensitive measurements, mode-locking in lasers, and phase-shift time-of-flight imaging (LiDAR). Here, we report a new type of single frequency intensity modulator that we refer to as a longitudinal piezoelectric resonant photoelastic modulator. The modulator consists of a thin lithium niobate wafer coated with transparent surface electrodes. One of the fundamental acoustic modes of the modulator is excited through the surface electrodes, confining an acoustic standing wave to the electrode region. The modulator is placed between optical polarizers; light propagating through the modulator and polarizers is intensity modulated with a wide acceptance angle and record breaking modulation efficiency in the megahertz frequency regime. As an illustration of the potential of our approach, we show that the proposed modulator can be integrated with a standard image sensor to effectively convert it into a time-of-flight imaging system.

[1] Department of Electrical Engineering, Stanford University, Stanford, CA 94305, USA. [2] Department of Applied Physics and Ginzton Laboratory, Stanford University, Stanford, CA 94305, USA. [3] Department of Microtechnology and Nanoscience (MC2), Chalmers University of Technology, Gothenburg, Sweden. ✉email: okan@stanford.edu

Controlling the intensity of a free-space optical beam is a problem of fundamental importance in optics. In this work, we focus on single-frequency free-space intensity modulators (SFFIM), a special type of intensity modulator that modulates the intensity of a free-space beam at a fixed frequency. The application spaces for these modulators include wide-field lock-in detection for sensitive measurements[1–5], mode-locking in lasers[6,7], and phase-shift time-of-flight imaging (LiDAR)[8–11]. It is desirable for an SFFIM to have low optical-insertion loss, high modulation efficiency, and a large acceptance angle[12,13]. The modulation frequency is application-specific, however, for many applications a higher frequency is desirable, especially for lock-in detection dominated by low-frequency noise and time-of-flight (ToF) imaging, where the ranging accuracy is proportional to the frequency of intensity modulation.

Achieving low optical-insertion loss, high modulation efficiency, and a large acceptance angle for an SFFIM at megahertz frequencies is a difficult and important challenge in optics. For modulation of light in the kilohertz frequency regime, liquid crystals offer good performance with low insertion losses and high modulation efficiencies[14–16]. Pockels cells (longitudinal and transverse), electroabsorption modulators using the quantum-confined Stark effect, and Bragg cells-based approaches allow modulation up to gigahertz frequencies. However, Bragg cells have a narrow acceptance angle[17], electroabsorption modulators have limited modulation efficiencies and face challenges if scaled to centimeter-square areas[9], and Pockels cells having high modulation efficiencies need centimeter thick, bulky crystals[18,19]. These bulky modulators are highly sensitive to environmental conditions (e.g., temperature variations). Transverse resonant photoelastic modulators, where the acoustic wave propagates perpendicularly to the optical wave, have an inherent trade-off between aperture size and modulation frequency—resulting in the aperture to be sub-millimeter for megahertz frequencies[20]. This renders them unsuitable for many applications[1–5,8–11]. Spatial light modulators (SLMs) infused with liquid crystals lack the speed to be operated at megahertz frequencies, and new types of high-speed SLMs have limited modulation efficiencies[21–23]. Metasurfaces have made significant progress over the last decade and have allowed unprecedented control over the properties of free-space beams[24–26]. However, active metasurfaces are still in their infancy and have limited functionality[27]. A better solution for free-space-optical modulation is urgently needed.

Here, we demonstrate a new type of intensity modulator that is compact, easy to manufacture, requires no DC bias, and has record modulation efficiency at megahertz frequencies. We demonstrate efficient intensity modulation at 3.7 MHz over a centimeter-square scale area, going beyond any other free-space intensity modulator. We also demonstrate efficient modulation for multiple angles of incidence. Illustrating the strength of our system, we experimentally show that the modulator can be used with a standard camera to enable high spatial-resolution ToF imaging. Our approach enables an attractive alternative to complex LiDAR systems relying on optical phased arrays (OPAs) and SLMs[22,28–32] with thousands of control elements or highly specialized costly image sensors that are difficult to implement with a large number of pixels[33–37].

## Results

**Modulator design**. Resonant designs are commonly used to improve the modulation efficiency of optical modulators operating at a single frequency. However, optical resonators with high quality factors come at the expense of a limited acceptance angle, while the attainable quality factors are limited for radio frequency (RF) resonators. To achieve both a high quality factor

and to break the trade-off between quality factor and acceptance angle (and therefore modulation efficiency), we choose to use an acoustic resonator. To construct an efficient intensity modulator incorporating an acoustic resonator and that is able to couple RF to optics, a material with suitable piezoelectric and photoelastic properties should be chosen. Here, we use lithium niobate (LN) as the modulator material, since it offers good piezoelectric and photoelastic properties, and is widely available at a low-cost (see Supplementary Information Section 9 for details on material choice).

We use a Y-cut LN wafer to break in-plane symmetry and allow the $x$ and $z$ directions to be modulated differently when an electric field is applied along the y direction of the wafer. To reach megahertz modulation frequencies over a centimeter-square area, the thickness of the wafer needs to be chosen such that the fundamental acoustic mode appears around megahertz frequencies. To satisfy this requirement, we use an LN wafer that is 0.5 mm thick with a diameter of 50.8 mm. The top and bottom surfaces of the wafer are coated with electrodes with a diameter of 12.7 mm. This allows us to reach a centimeter-square scale input aperture, while simultaneously confining the acoustic mode to the electrode region to limit clamping losses. The modulator is shown in Fig. 1a. The modulator has many acoustic modes that can be excited. These modes are shown in Fig. 1b, obtained by simulating the $|s_{11}|$ of the LN wafer using COMSOL[38].

The ideal acoustic mode has large piezoelectric and photoelastic couplings, as well as a uniform strain distribution in amplitude and phase in the electrode region. One such mode is the fundamental one shown in Fig. 1c—this is the most uniform acoustic mode that can be used for this modulator to achieve efficient modulation (see Supplementary Information Section 7 for some of the other acoustic modes that can be excited). We simulate the effect of a 2 Vpp signal applied to the surface electrodes using COMSOL at $f_c = 3.7696$ MHz (corresponding to the resonance in $|s_{11}|$ for Fig. 1c). The dominant strain $S_{yz}$ is shown in Fig. 1d. The contribution of the other strain components is negligible compared to $S_{yz}$ (see Supplementary Information Section 1 for details). Since an acoustic standing wave is excited, the phase of the strain is 0 or $\pi$ radians. The acoustic mode in Fig. 1d is composed of three different regions with roughly uniform strain amplitude, but with different phases (displayed as red and blue regions in Fig. 1d).

The strain in the wafer causes time-varying birefringence at the frequency $f_c$ of the signal applied to the electrodes. This is expressed in Eq. (1), where $n_o$ and $n_e$ are the ordinary and extraordinary refractive indices of LN, $p_{14}$ is the photoelastic constant of LN which relates the volume average shear strain $\bar{S}_{yz} = A\cos(2\pi f_c t)$ to the index ellipsoid of LN. $n_x(t)$, $n_y(t)$, and $n_z(t)$ are the modulated refractive indices of the LN wafer. The volume average shear strain magnitude is $A = 1.00 \times 10^{-5}$ for the COMSOL simulation. This volume average strain is calculated for a 1 cm diameter region centered on the wafer and over a thickness of 0.5 mm.

$$\frac{1}{n_x^2(t)} = \frac{1}{n_o^2} + 2p_{14}\bar{S}_{yz}$$
$$\frac{1}{n_y^2(t)} = \frac{1}{n_o^2} - 2p_{14}\bar{S}_{yz} \qquad (1)$$
$$\frac{1}{n_z^2(t)} = \frac{1}{n_e^2}$$

A laser beam propagating through the transparent electrode region excites an ordinary and an extraordinary wave in the wafer (due to anisotropy of LN), and these two waves pick up time-varying phases $\phi_o(t)$ and $\phi_e(t)$ upon propagation through the wafer, respectively. This results in the polarization of the laser beam that has propagated through the wafer to rotate in time with frequency $f_c$.

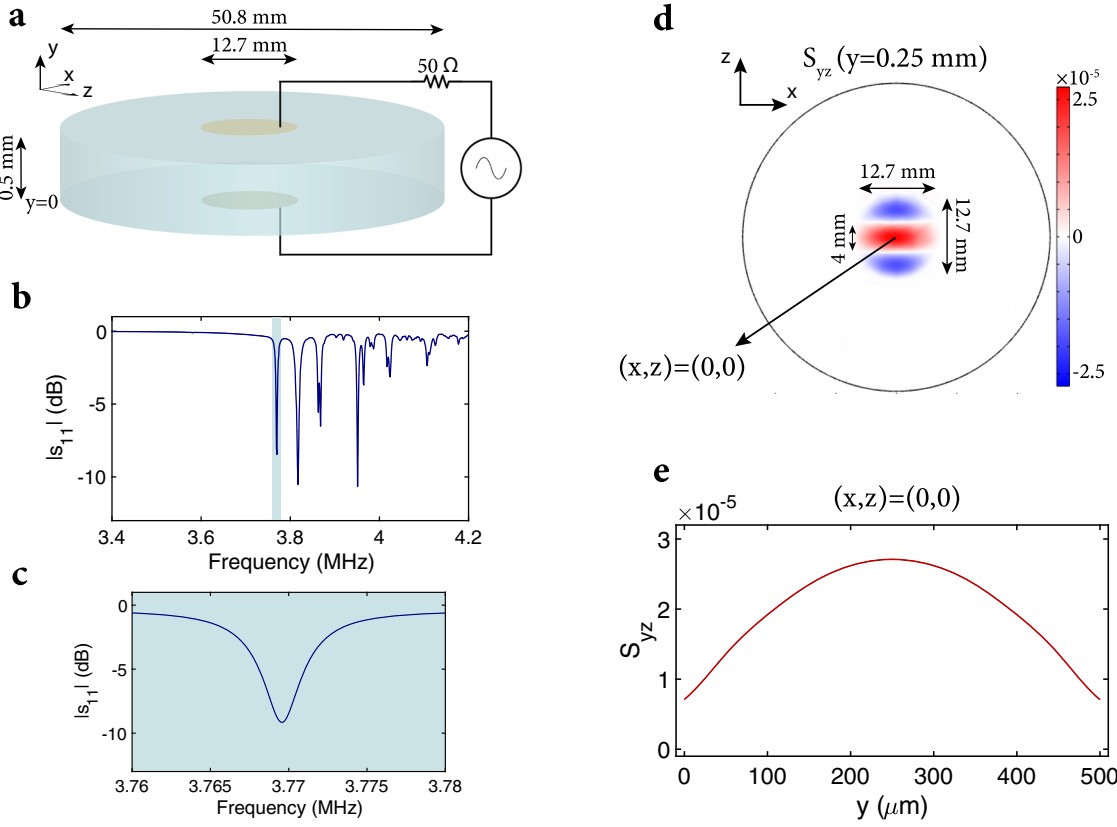

**Fig. 1 Exciting the correct acoustic mode of the wafer. a** A Y-cut lithium niobate wafer of diameter 50.8 mm and of thickness 0.5 mm is coated on top and bottom surfaces with electrodes having a diameter of 12.7 mm. The wafer is excited with an RF source through the top and bottom electrodes. **b** Simulated $|s_{11}|$ of the wafer with respect to 50 Ω, showing the resonances corresponding to different acoustic modes of the wafer (loss was added to lithium niobate to make it consistent with experimental results). The desired acoustic mode appears around 3.77 MHz and is highlighted in blue. **c** The desired acoustic mode $|s_{11}|$ with respect to 50 Ω is shown in more detail. **d** The dominant strain distribution ($S_{yz}$) when the wafer is excited at 3.7696 MHz with 2 Vpp is shown for the center of the wafer. This strain distribution corresponds to the $|s_{11}|$ resonance shown in (**c**). **e** The variation in $S_{yz}$ parallel to the wafer normal and centered along the wafer is shown when the wafer is excited at 3.7696 MHz with 2 Vpp.

We place the wafer between optical polarizers with transmission axis $\hat{\mathbf{t}} = (\hat{\mathbf{a}}_x + \hat{\mathbf{a}}_z)/\sqrt{2}$ to convert polarization modulation into intensity modulation. The intensity $I(t)$ of a plane wave with random polarization that has passed through the three components is expressed in Eq. (2) (see Supplementary Information Section 2 for derivation), where $I_0$ is the intensity of the incoming plane wave, $c_o$ is the amplitude of the excited ordinary wave, $c_e$ is the amplitude of the excited extraordinary wave, HOH stands for the higher order harmonics, $J_0$ and $J_1$ stand for the zeroth and first-order Bessel functions of the first kind, respectively. The static phase accumulation $\phi_s$ and the dynamic phase accumulation $\phi_D$ are found as follows: $\phi_s + \phi_D \cos(2\pi f_c t) = \phi_o(t) - \phi_e(t)$.

$$I(t) = \frac{I_0}{2}\left(c_o^4 + c_e^4 + 2c_o^2 c_e^2 \left[\cos(\phi_s)\left(J_0(\phi_D)\right.\right.\right. \\ \left.\left.\left. - 2\sin(\phi_s)J_1(\phi_D)\cos(2\pi f_c t) + \text{HOH}\right]\right) \quad (2)$$

**Experiments**. To demonstrate intensity modulation using this approach, we coat the top and bottom surfaces of a double-side polished 50.8 mm diameter and 0.5-mm-thick Y-cut LN wafer with indium tin oxide (ITO) to serve as transparent surface electrodes. The desired acoustic mode is found by measuring the reflection scattering parameter $s_{11}$ using a vector network analyzer (VNA). The dips in $|s_{11}|$ occur at similar frequencies as in the COMSOL simulation shown in Fig. 1c, with the fundamental appearing around 3.73 MHz. The measured $|s_{11}|$ for the desired acoustic mode of the wafer is shown in Fig. 2c, with a quality

factor of around $10^3$. To measure the intensity modulation efficiency and to verify that the acoustic mode profile matches simulation (Fig. 1d), the coated wafer is placed between optical polarizers, and a laser beam with wavelength 532 nm is passed through the polarizer, wafer electrode region, and polarizer, as shown in Fig. 2a. 90 mW of RF power is applied to the surface electrodes at $f_r = 3.7337$ MHz, which corresponds to the resonance observed in $|s_{11}|$ in Fig. 2c. 90 mW of RF power corresponds to 6 Vpp over the wafer surface electrodes.

To detect the intensity modulation profile of the laser beam with high spatial resolution, we use a standard CMOS camera offering four-megapixel resolution. Since the frame rate of the camera is limited to less than a kilohertz, and the modulator operates at $f_r = 3.7337$ MHz, we intensity modulate the laser beam at $f_r + 4$ Hz by using a free-space acousto-optic modulator (see "Methods" for details) to perform heterodyne detection and measure the 4 Hz beat tone using the camera, as shown in Fig. 2a. The depth of intensity modulation is proportional to the amplitude of the beat tone, and its phase is related to the acoustic standing wave phase. The laser beam intensity profile, the normalized amplitude of the beat tone (depth of modulation), and the phase of the beat tone are shown in Fig. 2f, g, h, respectively. The inferred volume average strain over 1-cm diameter region centered on the wafer and over a thickness of 0.5 mm for the experiment is $4.15 \times 10^{-5}$ (see Supplementary Information Section 4 for details). This agrees relatively well with the COMSOL simulation ($1.00 \times 10^{-5} \times \frac{6\,\text{Vpp}}{2\,\text{Vpp}} = 3.00 \times 10^{-5}$). The

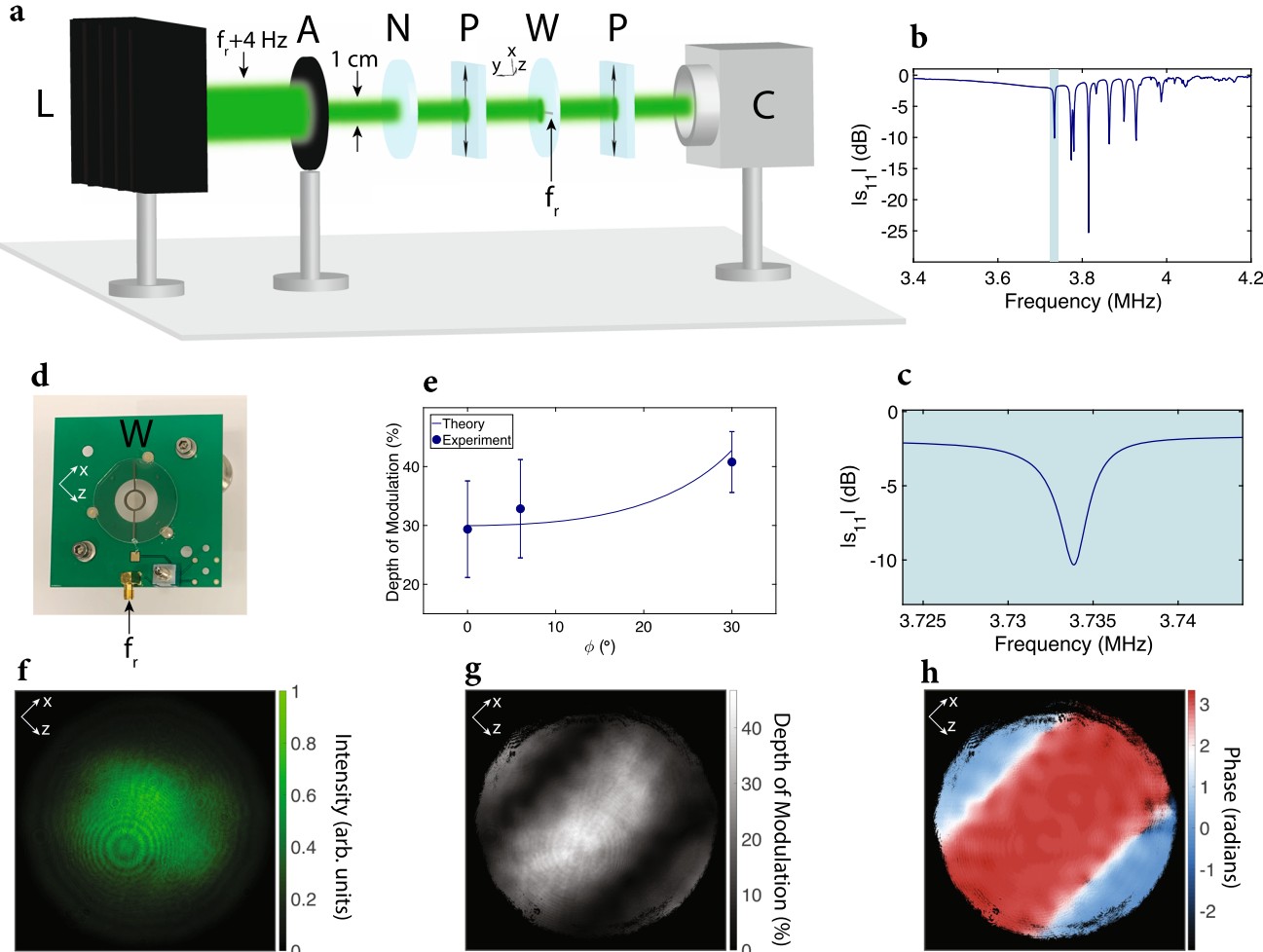

**Fig. 2 Experimental characterization of the modulator. a** Schematic of the characterization setup is shown. The setup includes a laser (L) with a wavelength of 532 nm that is intensity-modulated at 3.733704 MHz, aperture (A) with a diameter of 1 cm, neutral density filter (N), two polarizers (P) with transmission axis $\hat{\mathbf{t}} = (\hat{a}_x + \hat{a}_z)/\sqrt{2}$, wafer (W), and a standard CMOS camera (C). The wafer is excited with 90 mW of RF power at $f_r = 3.7337$ MHz, and the laser beam passes through the center of the wafer that is coated with ITO. The camera detects the intensity-modulated laser beam. **b** The desired acoustic mode is found for the modulator by performing an $s_{11}$ scan with respect to 50 Ω using 0 dBm excitation power and with a bandwidth of 100 Hz. The desired acoustic mode is highlighted in blue. **c** The desired acoustic mode is shown in more detail by performing an $s_{11}$ scan with respect to 50 Ω using 0 dBm excitation power with a bandwidth of 20 Hz. **d** The fabricated modulator is shown. **e** The depth of intensity modulation is plotted for different angles of incidence for the laser beam (averaged across all the pixels), where $\phi$ is the angle between the surface normal of the wafer and the beam direction $\hat{\mathbf{k}}$ (see "Methods" for more details). Error bars represent the standard deviation of the depth of intensity modulation across the pixels. **f** Time-averaged intensity profile of the laser beam detected by the camera is shown for $\phi = 0$. **g** The DoM at 4 Hz of the laser beam is shown per pixel for $\phi = 0$. **h** The phase of intensity modulation at 4 Hz of the laser beam is shown per pixel for $\phi = 0$.

ability to modulate different angles of incidence is shown in Fig. 2e, where $\phi$ is the angle between the wafer normal and the laser beam propagation direction (see "Methods" for more details).

We use the fabricated modulator for phase-shift-based ToF imaging[35]. In this imaging modality, intensity-modulated light is used to illuminate targets in a scene. The targets at different distances to the transmitter reflect back to the receiver with different phase shifts, where the phase shift $\Phi$ in radians is related to the distance $d$, speed of light in air $c$, and the intensity modulation frequency $f_m$ as shown in Eq. (3).

$$d = \frac{\Phi c}{4\pi f_m} \tag{3}$$

Higher modulation frequencies are favorable due to providing better range resolution. Megahertz frequencies are usually used since the unambiguous imaging range of $c/(2f_m)$ due to phase

wrapping is on the meters level, while also offering good range resolution. Since standard cameras have frame rates limited to less than kilohertz frequencies, they cannot sample the megahertz modulation frequency. The modulator allows the phase of the megahertz signal to be detected by the camera through heterodyne detection.

The ToF imaging setup using a standard camera and the modulator is shown in Fig. 3a. A laser of wavelength 635 nm is intensity-modulated at 3.733702 MHz and is used to illuminate two targets. We place the modulator between two optical polarizers, and we place an aperture in front of the wafer to only use the center 4 mm diameter region so that destructive interference between the anti-phase regions is avoided. We attach a camera lens to the camera to resolve the two targets. Applying Eq. (3) to the captured frames having a beat tone at 2 Hz, the distance that each camera pixel corresponds to is calculated, and the reconstructed depth map is shown in Fig. 3c. The target

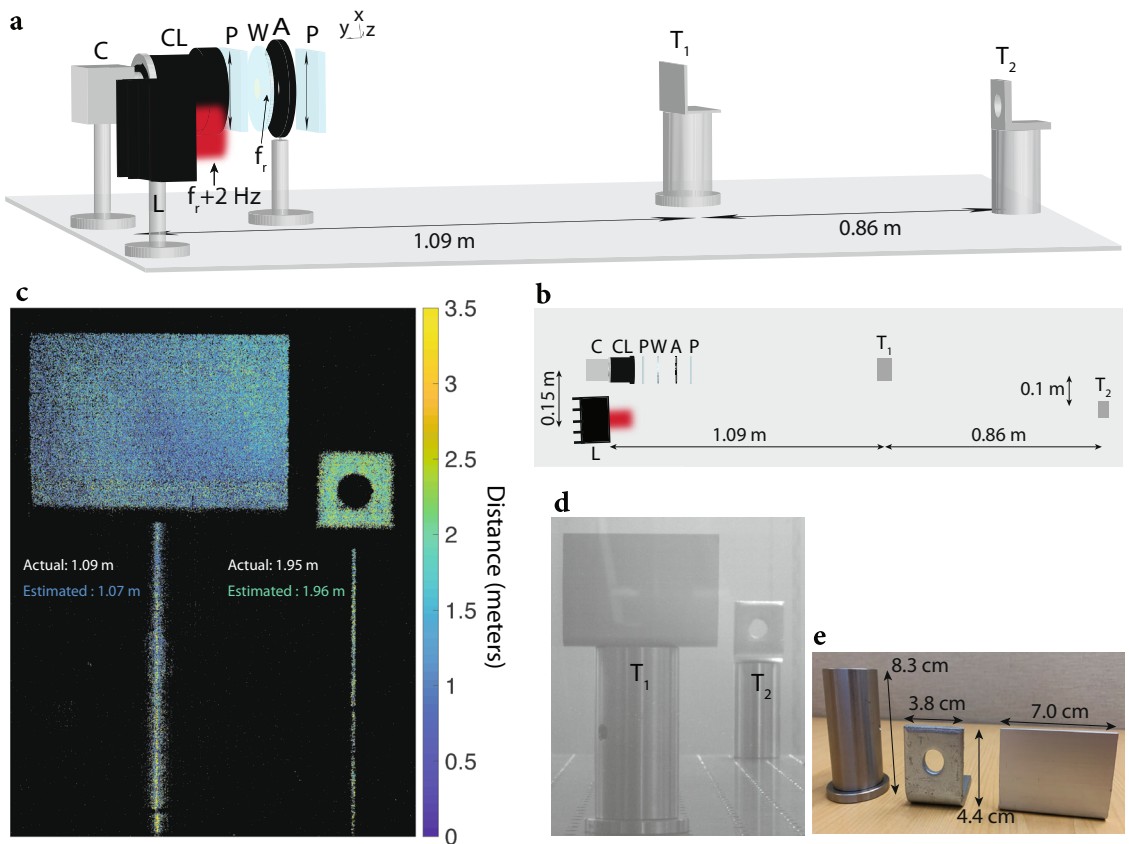

**Fig. 3 ToF imaging using the modulator and a standard camera. a** Schematic of the imaging setup is shown. The setup includes a standard CMOS camera (C), camera lens (CL), two polarizers (P) with transmission axis $\hat{\mathbf{t}} = (\hat{a}_x + \hat{a}_z)/\sqrt{2}$, wafer (W), aperture (A) with a diameter of 4 mm, laser (L) with a wavelength of 635 nm that is intensity-modulated at 3.733702 MHz, and two metallic targets ($T_1$ and $T_2$) placed 1.09 m and 1.95 m away from the imaging system, respectively. For the experiment, 140 mW of RF power at $f_r = 3.7337$ MHz is used to excite the wafer electrodes. The laser is used for illuminating the targets. The camera detects the reflected laser beam from the two targets, and uses the 2 Hz beat tone to extract the distance of each pixel corresponding to a distinct point in the scene (see "Methods" for more details). **b** Bird's eye view of the schematic in (**a**). **c** Reconstructed depth map seen by the camera. Reconstruction is performed by mapping the phase of the beat tone at 2 Hz to distance using Eq. (3). The distance of each pixel is color-coded from 0 to 3 m (pixels that receive very few photons are displayed in black). The distance of targets $T_1$ and $T_2$ are estimated by averaging across their corresponding pixels, respectively. The estimated distances for $T_1$ and $T_2$ are 1.07 m and 1.96 m, respectively (averaged across all pixels corresponding to $T_1$ and $T_2$). **d** Ambient image capture of the field-of-view of the camera, showing the two targets $T_1$ and $T_2$. **e** The dimensions of the targets used for ToF imaging are shown.

distances are reconstructed to centimeter-level accuracy. The reason why only a thin slice of the optical posts are seen in the depth reconstruction for Fig. 3c is due to specular reflection from the shiny metal posts. This is a common problem for all LiDAR and optical ToF imaging systems[39,40].

## Discussion

The experiment shows that high spatial-resolution ToF imaging can be achieved with a standard camera and the modulator. It also demonstrates the optically broadband nature of the modulator, being able to modulate multiple different wavelengths (limited by the transparency window of LN and ITO). The high spatial resolution for the depth reconstruction is enabled by the megapixel resolution of the camera. The imaging performance could be significantly improved by depositing a static polarization manipulating metasurface[41] on the electrodes of the wafer. Such a metasurface could allow the whole aperture to be utilized (compensating for the phase of the anti-phase regions shown in Fig. 1d by controlling the sign of $\sin \phi_s$), and also allow the modulation efficiency to remain high for a large acceptance range by canceling the static phase variation $\phi_s$ for different angles of incidence.

A comparison table is provided in Table 1 for different modulation approaches. The two performance metrics used are optical-insertion loss and the RF power required to achieve 100% intensity modulation of the laser beam. Only modulators that can achieve large acceptance angles are included. For the Pockels cell approaches, a thickness of 0.5 mm along the light propagation direction is assumed to have the same temperature tolerance as reported in this work. The 4.2 dB optical-insertion loss reported could be brought closer to 3 dB (due to polarizers) by depositing anti-reflection coatings on the wafer. We have neglected cut-off frequency issues due to capacitance and have only focused on efficiency. The modulators compared to in Table 1 are capacitive and will face challenges when scaled to centimeter-square areas without matching circuits. The impedance of our modulator is well matched to 50 Ω and therefore does not require a matching circuit. In fact, the 7.4 W/cm² reported is a worst-case for our device, since more energy is required to achieve 100% modulation due to the Bessel function compared to other modulation approaches (see Supplementary Information Section 6 for more details). The modulation efficiency of our approach could be further improved by investigating hybrid platforms including multiple layers of materials to separate the piezoelectric and

**Table 1 Comparison table.**

| Modulation principle | Optical wavelength (nm) | Optical insertion loss (dB) | RF power required to achieve 100% intensity modulation (W/cm$^2$) | Optically broadband |
|---|---|---|---|---|
| Transverse resonant Pockels[18] | 532 | 3 | $6.7 \times 10^5$ | Yes |
| Longitudinal Pockels[19] | 532 | 3 | $1.83 \times 10^4$ | Yes |
| Electroabsorption[9] | 860 | – | 34 | No |
| Gate-tunable metasurface[23] | 1550 | – | 460 | Yes |
| Plasmonic nanoresonator[22] | 1340 | 20 | 454 | Yes |
| Photoelastic (this work) | 532 | 4.2 | 7.4 | Yes |

Modulator performances are compared in terms of intensity modulation at 3.7 MHz. The two performance metrics used are the optical-insertion loss and the RF power required to achieve 100% intensity modulation of a laser beam. To show modulation efficiency, the RF power required to achieve 100% intensity modulation is calculated by extrapolating the values reported in the references.

photoelastic functionalities to different materials. These platforms could offer higher acoustic quality factors for some acoustic modes and therefore improve the modulation efficiency.

To conclude, we have demonstrated a new resonant free-space intensity modulator that modulates light from visible and up to near-infrared wavelengths at megahertz frequencies with record efficiency. The modulator can find immediate use in applications requiring free-space beams to be intensity-modulated with low RF power at megahertz frequencies over centimeter-square-scale apertures. It could enable low-cost and high spatial-resolution ToF imaging and LiDAR with low-cost standard image sensors.

## Methods

**Modulator fabrication**. We fabricate the piezoelectric-resonant photoelastic modulator using a 50.8 mm diameter, 0.5-mm thick double-side polished Y-cut LN wafer. A metal plate with dimensions 63.5 mm by 63.5 mm and thickness 1 mm is drilled in the center with a hole diameter of 12.7 mm. The metal plate is centered and placed on top of the wafer and approximately 450-nm thick ITO is deposited using sputter coating in a load-locked chamber. This sputtering process using the metal plate is repeated for the backside of the wafer. During the sputtering process, the chamber is not heated to prevent the formation of black LN. The optical transmission of ITO coated on the LN wafer is improved by heating on a hot plate for 26 min at a temperature of 250 °C. The sheet resistance of ITO after heating is ~25 Ω/sq.

A metal plate with dimensions 63.5 mm by 63.5 mm and thickness 1 mm is drilled such that the metal region that lies between two circles centered on the plate with diameter 10.16 mm and 12.7 mm is removed. A rectangular region that is 6.35 mm away from the center of the plate and with dimensions 1 mm by 16.5 mm is also drilled. The metal plate is centered and placed on top of the wafer and 300-nm thick aluminum is evaporated in a load-locked chamber. This evaporation process using the metal plate is repeated for the backside of the wafer.

The ITO and aluminum-coated wafer is attached to a PCB by gluing using epoxy to three plastic washers. The top surface electrode is electrically connected to the PCB signal port using two gold wirebonds, and the bottom surface electrode is electrically connected to the PCB ground using two gold wirebonds. The wirebonds for the top and bottom surfaces of the wafer connect to the microstrip aluminum region defined by the rectangular region (dimensions 1 mm by 16.5 mm) drilled on the metal plate.

**Experimental setup for modulator characterization**. For RF characterization of the modulator, we perform an $s_{11}$ scan with respect to 50 Ω using a VNA (Roh-de&Schwarz ZNB20) with excitation power of 0 dBm and bandwidth of 100 Hz for the broad scan with a frequency step size of 100 Hz (Fig. 2b) and a bandwidth of 20 Hz and frequency step size of 10 Hz for the focused scan (Fig. 2c), respectively. The modulator is excited through the SMA connector attached to the PCB holding the modulator (Fig. 2d).

For optical characterization of the modulator, we use the setup shown in Fig. 2a. In this setup, a diode-pumped solid-state (DPSS) laser diode of wavelength 532 nm (Thorlabs DJ532-10) mounted on a laser mount (Thorlabs TCLDM9) is used. The laser beam is intensity-modulated using a free-space acousto-optic modulator (G&H AOMO 3080-125). The acousto-optic modulator (AOM) is excited at its center frequency of 80 MHz, and this carrier frequency is intensity-modulated at 3.733704 MHz. This causes the light beam passing through the AOM to have diffracted beams, and these beams to be intensity modulated at 3.733704 MHz. We use the first-order beam for characterizing the modulator. This beam is passed through an iris (Thorlabs ID25) with diameter adjusted to 1 cm to match the beam diameter to the active region of the modulator (ITO-covered region).

The intensity-modulated laser beam passes through a variable neutral density filter (OptoSigma NDHN), a wire-grid polarizer (Thorlabs WP25L-VIS), the center

of the modulator (the modulator is excited with 90 mW of power at 3.7337 MHz through the SMA connector of the PCB it is attached to), and then through another wire-grid polarizer. The signal generators driving the AOM and the proposed modulator are synchronized. The two polarizers and the wafer surfaces are placed parallel to each other and the centers of the three components are aligned. The wafer optic axis makes a 45° angle with the transmission axis of the two polarizers. The laser beam that has passed through the first polarizer, the wafer, and the second polarizer (analyzer) is detected with a standard camera (Basler acA2040-90um) using a frame rate of 30 Hz, 16 bit precision per pixel, 400 µs exposure time; 600 frames are captured. The depth of intensity modulation is calculated by performing a time-domain fast Fourier transform (FFT) on each pixel of the camera, and finding the ratio of the beat tone at 4 Hz multiplied by four to the DC level. The phase of modulation is calculated by finding the phase of the beat tone at 4 Hz after performing the FFT on each pixel.

The intensity modulation using the AOM does not result in a homogeneous modulation of the laser beam. Since the acoustic wave is turned on and off at a frequency of 3.733704 MHz, the phase of the intensity modulation for the laser beam has a linear phase variation depending on the laser beam size and the acoustic wavelength corresponding to 3.733704 MHz in the AOM. This linear phase variation of the intensity modulation frequency of 3.733704 MHz for the laser beam varies along the direction of acoustic wave propagation in the AOM. This phase variation is removed in Fig. 2h by a least-squares fit to the phase of modulation calculated after performing the FFT.

Depth of intensity modulation for different angles of incidence of the laser beam to the wafer surface is also captured. The angle of incidence of the laser beam is such that it makes an equal angle with the x and z axes of the LN wafer. In standard spherical coordinate notation, the two angles used to describe the laser beam

direction $\hat{\mathbf{k}}$ can be expressed as $\theta = \cos^{-1}\left(\frac{\sin\phi}{\sqrt{2}}\right)$ and $\psi = \sin^{-1}\left(\frac{\cos\phi}{\sin\left(\cos^{-1}\left(\frac{\sin\phi}{\sqrt{2}}\right)\right)}\right)$.

The depth of intensity modulation is measured as described in the previous paragraph for each of the different angles of incidence. The values reported in Fig. 2e are the mean DoM values of the pixels in the center region lying between the two null regions.

The optical-insertion loss of the wafer is determined by passing the 532-nm laser beam through the ITO-coated region of the wafer and measuring the DC level with a photodetector (Thorlabs DET36A2). The optical insertion loss is found to be 1.2 dB, calculated by taking the ratio of the light intensity measured by the photodetector with and without the wafer placed. The optical-insertion loss of 4.2 dB reported in Table 1 also includes the 3 dB loss of the polarizer when unpolarized light is assumed.

**Experimental setup for time-of-flight imaging**. A laser diode of wavelength 635 nm (Thorlabs HL6322G) and with output optical power less than 15 mW is mounted on a laser mount (Thorlabs TCLDM9) and intensity-modulated at 3.733702 MHz using a signal generator. The laser diode has a divergence of 8° by 30° (FWHM). The depth of intensity modulation is approximately 100% for the laser diode, measured with a photodetector (Thorlabs DET36A2). The intensity-modulated laser beam is used to illuminate targets $T_1$ and $T_2$ shown in Fig. 3. The reason why the modulator characterization is done using a DPSS laser with a wavelength of 532 nm is because of the stability and low divergence angle of the laser beam. The stable laser beam with a low divergence angle from the DPSS laser allows the optical characterization to be performed. Modulating the DPSS laser through current modulation is limited to low frequencies (below 1 MHz). That is why an AOM was used to modulate the laser beam emitted by the DPSS laser. Intensity modulation using an AOM is avoided for ToF imaging due to the linear phase variation of the intensity modulation frequency across the laser beam (as described in the previous section). Since distance is encoded in the phase of the beat frequency, using an AOM for intensity modulation would make the depth map reconstruction difficult. This is the reason why a laser diode with a wavelength

of 635 nm and a larger divergence angle was used for performing the ToF imaging experiment.

For the ToF imaging setup, we place the wafer between two optical polarizers such that the transmission axis of the two polarizers makes a 45° angle with the optical axis of the wafer, and the wafer and polarizer surfaces are placed parallel to each other. An iris (Thorlabs ID25) with diameter adjusted to 4 mm is placed between the polarizer and wafer such that the laser beam only passes through the center 4 mm of the wafer. This causes the laser beam to be approximately uniformly modulated by preventing the beam from being modulated by the anti-phase regions (see Fig. 1d). The modulator is excited with 140 mW of power at 3.7337 MHz through the SMA connector of the PCB. The four components (polarizer, modulator, iris, polarizer) are placed in front of a standard camera (Basler acA2040-90um). The signal generators driving the laser diode and the modulator are synchronized.

The reflected light from the two targets is captured by the camera. A camera lens with a diameter of 45.0 mm (Kowa LM75HC) is attached to the camera to resolve the two targets on the camera image sensor. The camera is operated with a frame rate of 10 Hz, 16 bit precision per pixel, exposure time of 99 ms, and an internal gain of 23 dB; 600 frames are captured. The distance that each pixel corresponds to is calculated using Eq. (3). The resulting depth map is shown in Fig. 3c. The standard deviation for the depth map in Fig. 3c is 1.0 m per pixel for the pixels corresponding to $T_1$, and 0.92 m per pixel for the pixels corresponding to $T_2$. For capturing the ambient image (Fig. 3d), 16 bit precision is used with 1 s of capture time and 11 dB internal gain for the camera.

## Data availability
Data that support the findings of this study are available in the main text and Supplementary Information. Data of this study are also available at https://github.com/okanatalar/piezoelectric-resonant-photoelastic-modulator and from the corresponding author upon reasonable request.

## Code availability
The codes used for this study are available at https://github.com/okanatalar/piezoelectric-resonant-photoelastic modulator and from the corresponding author upon reasonable request.

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

## Acknowledgements
The authors thank Christopher J. Sarabalis for useful discussions. This work was funded in part by Stanford SystemX Alliance, Office of Naval Research, and NSF ECCS-1808100.

## Author contributions
O.A. performed the fabrication and experiments. O.A., A.H.S.-N., R.V.L., and A.A. conceived the idea, analyzed the data, and wrote the paper. A.H.S.-N. and A.A. supervised the project.

## Competing interests
O.A., A.H.S.-N., and A.A. are inventors of US patent application 16/971,127. R.V.L. declares no competing interests.
