## [Peer Review File · Nature Communications]

REVIEWER COMMENTS

Reviewer #1 (Remarks to the Author):

Atalar and coworkers report on a piezo-electrically driven photoelastic modulator device for LiDAR applications.

In my view, the manuscript certainly has the potential to be publishable in a journal the caliber of Nature Communications based on the following reasons:

To my knowledge, this type of device is novel and goes beyond the state of the art. Also, the type of modulation mechanism has not been employed at least not for this application.

Comparison with literature on competitor LiDAR systems based on other tuning mechanisms, the presented device offers key advantages, in particular operation at moderate (lower) power levels.

The device is made of Lithium Niobate, an established platform for microacoustics and (nonlinear) photonics. This makes the paper very appealing since it promises swift raising of the TRL.

Another strong point is that the authors demonstrate a relatively straightforward integration of the device in a prototype setup in which ToF LiDAR is implemented by a standard CMOS camera.

Overall, the paper is technically sound, the presented data, their analysis and interpretation are convincing apart from a few minor questions listed below.

These questions are as follows:

(1) The rf characterization data in Figure 1b, a number of modes are resolved. The authors selected the lowest frequency mode for operation and state that the others are not suitable because they cannot achieve "homogeneous" modulation. Here more information is necessary. It is not clear to me if this is related to the spatial profile of the modes themselves or if it related to the inhomogeneities of the substrate (or a combination or other reasons). This point has to be addressed and clarified.

(2) The authors perform an elaborate study of the spatial strain profiles across the area of the device. It is not clear (at least not to me) if these types of studies would have to be performed for every single device fabricated because of the inherent limitations of the LNO substrate. If so, this would significantly hamper raising the TRL. Please comment.

(3) The device is fabricated on LNO for good reasons. There are other candidate piezoelectrics e.g. established materials like quartz or Al(Sc)N or electro-optical active ones e.g. BTO. The comparisons made should be extended from operation principles to classes of materials to make an ever stronger point here.

With these points addressed the manuscript could be considered further for publication in Nature Communications.

Reviewer #2 (Remarks to the Author):

The manuscript reports on a photoelastic modulator based on piezoelectrically driven bulk acoustic waves (BAWs) propagating along the Y direction of a Y-cut LiNbO₃ wafer. In the modulator, acoustic vibrations with a strong shear component and half-wavelength approximately equal to the wafer thickness are piezoelectric excited via semi-transparent contacts deposited on opposite surfaces of the wafer. The LiNbO₃ crystal is optically anisotropic in the plane perpendicular to the BAW propagation. The vibrations modulate in anti-phase the dielectric properties along two orthogonal directions on this plane at the BAW frequency, thus turning the structure into a time-dependent optical retarder with retardation controlled by the BAW amplitude. By using polarizers, the modulation of the retardation can be turned into a modulation of the transmitted intensity. The main advantages of the modulation concept are, apart from the simple design, the

broad wavelength range, the high modulation efficiency, and the wide optical acceptance angle. As an example of application, the modulator is used for time-of-flight imaging with cm resolution.

The experimental work as well as the extensive modelling and simulations used for data interpretation fully support the conclusions drawn in the manuscript. Also, the authors give an illustrative application example and also compare their approach with competing ones. The principles of operation of the proposed modulator (based on conventional acousto-optics) are, however, very well established. Although the device offers advantages regarding some properties (e.g., wavelength range and acceptance angles), the operation frequencies are rather modest (few MHz, depending on crystal thickness) and cannot be easily scaled up, thus implying, e.g., in low spatial resolution for time-of-flight imaging). Furthermore, the integration with other functionalities is hindered by the need of crystals with well-defined orientations. The results are certainly of immediate interest for the applied acousto-optic community, they lack, however, the degree of novelty and generality for a wide scientific community. For this reason, I do not recommend publication in Nat. Communication.

Other suggestions/comments:

- The phase modulation along the z-direction in Fig. 1d severely restricts the useful aperture of the modulator. The authors mention that this drawback can be lifted by "depositing a static polarization manipulating metasurface" (line 266). Since the changes are dynamic, it is not clear how the metasurface can compensate for the time-dependent anti-phase ranges shown in Fig. 1d (unless, of course, the strain-dependence of the metasurface polarization is the same as in the bulk, which is very unlikely).
- Line 646-647: I guess that the standard deviation in Fig. 3c is probably 1 cm (and not 1m, as listed in the text).

Reply to Reviewers

We thank the reviewers and the editor for insightful comments and detailed feedback. We have modified the manuscript to address the issues. We have used the comments of the reviewers to improve the paper and better convey the concepts to readers.

Key to the responses:

- Reviewer comments are in black
- Response to reviewer comments are in blue
- Text from the modified manuscript is in green

REVIEWER COMMENTS

Reviewer #1 (Remarks to the Author):

Atalar and coworkers report on a piezo-electrically driven photoelastic modulator device for LiDAR applications.

In my view, the manuscript certainly has the potential to be publishable in a journal the caliber of Nature Communications based on the following reasons:

To my knowledge, this type of device is novel and goes beyond the state of the art. Also, the type of modulation mechanism has not been employed at least not for this application.

Comparision with literature on competitor LiDAR systems based on other tuning mechnisms, the presented device offers key advantages, in particular operation at moderate (lower) power levels.

The device is made of Lithium Niobate, an established platform for microacoustics and (nonlinear) photonics. This makes the paper very appealing since it promises swift raising of the TRL.

Another strong point is that the authors demonstrate a relatively straightforward integration of the device in a prototype setup in which ToF LiDAR is implemented by a standard CMOS camera.

Overall, the paper is technically sound, the presented data, their analysis and interpretation are convincing apart from a few minor questions listed below.

- We thank the reviewer for their support and interest in our work and for the valuable comments.

These questions are as follows:

(1) The rf characterization data in Figure 1b, a number of modes are resolved. The authors selected the lowest frequency mode for operation and state that the others are not suitable because they cannot achieve “homogeneous” modulation. Here more information is necessary. It is not clear to me if this is related to the spatial profile of the modes themselves or if it related to the inhomogeneities of the substrate (or a combination of other reasons). This point has to be addressed and clarified.

- This is an important point raised by the reviewer. This is related to the spatial profile of the acoustic modes, and not to the inhomogeneities in the substrate. Figure below shows some of the other modes (for the dominant S_{yz} shear strain) that can be excited by the surface electrodes. Most of the acoustic modes have many nulls, making the modulator more difficult to use for homogenous intensity modulation purposes.

Fig. S9. Strain profile in the wafer for other acoustic modes. **a** S_{yz} strain profile for the plane that is parallel and 0.25 mm above the bottom wafer surface. 2Vpp is applied to wafer surface electrodes at a frequency of 3.817 MHz. **b** Simulated $|s_{11}|$ of the wafer with respect to 50 Ω , with the light blue region showing the excited mode in (a). **c** S_{yz} strain profile for the plane that is parallel and 0.25 mm above the bottom wafer surface. 2Vpp is applied to wafer surface electrodes at a frequency of 3.868 MHz. **d** Simulated $|s_{11}|$ of the wafer with respect to 50 Ω , with the light blue region showing the excited mode in (c). **e** S_{yz} strain profile for the plane that is parallel and 0.25 mm above the bottom wafer surface. 2Vpp is applied to wafer surface electrodes at a frequency of 3.951 MHz. **f** Simulated $|s_{11}|$ of the wafer with respect to 50 Ω , with the light blue region showing the excited mode in (e).

- The main text in the manuscript is modified to make this point more clear (Starting at line 109): “this is the most uniform acoustic mode that can be used for this modulator to achieve efficient modulation (see Supplementary Information Section 7 for some of the other acoustic modes that can be excited).”

- A new supplementary information section has been added to the manuscript to make this point more clear in the text - Supplementary Information 7, along with a new figure - Fig. S9.

(2) The authors perform an elaborate study of the spatial strain profiles across the area of the device. It is not clear (at least not to me) if these types of studies would have to be performed for every single device fabricated because of the inherent limitations of the LNO substrate. If so, this would significantly hamper raising the TRL. Please comment.

- The device parameters are affected by the wafer thickness, electrode diameter, and quality factor (due to limitations on the purity of LNO). Below shown are simulations for different thicknesses, quality factor, and electrode area - Fig. S10. The acoustic mode used (its spatial distribution), as seen in Fig. S10, is highly insensitive to these variations. The device just needs to be excited at a slightly different frequency based on the wafer thickness, which can easily be achieved by performing an s_{11} measurement after the device is fabricated to check its resonance frequency.

Line 222 in Supplementary Information - “ The acoustic mode is tolerant to small variations in these parameters (deviations by less than 10%). The acoustic resonance frequency varies with the wafer thickness; the mechanical equivalent resistance for the acoustic mode varies with the electrode diameter and the acoustic quality factor.”

- This missing piece is now added to supplementary information as a new section - Supplementary Information 8, along with simulations assuming 10% tolerance for the uncertainties.

Fig. S10. Sensitivity analysis for the acoustic mode. **a** S_{yz} strain profile for the plane that is parallel and 0.25 mm above the bottom wafer surface. 2Vpp is applied to wafer surface electrodes at a frequency of 3.554 MHz. Wafer thickness is 530 μm , electrode diameter is 13.3 mm, and acoustic quality factor is 900. **b** Simulated $|s_{11}|$ of the wafer described in **(a)**. **c** S_{yz} strain profile for the plane that is parallel and 0.25 mm above the bottom wafer surface. 2Vpp is applied to wafer surface electrodes at a frequency of 3.698 MHz. Wafer thickness is 510 μm , electrode diameter is 12.3 mm, and acoustic quality factor is 1,100. **d** Simulated $|s_{11}|$ of the wafer described in **(c)**. **e** S_{yz} strain profile for the plane that is parallel and 0.25 mm above the bottom wafer surface. 2Vpp is applied to wafer surface electrodes at a frequency of 3.974 MHz. Wafer thickness is 475 μm , electrode diameter is 11.4 mm, and acoustic quality factor is 1,000. **f** Simulated $|s_{11}|$ of the wafer described in **(e)**.

(3) The device is fabricated on LNO for good reasons. There are other candidate piezoelectrics e.g. established materials like quartz or Al(Sc)N or electro-optical active ones e.g. BTO. The comparisons made should be extended from operation principles to classes of materials to make an ever stronger point here.

- The reviewer brings up a very important point. Lithium niobate is the material of choice for the modulator due to low acoustic loss (translating to high attainable quality factor, and therefore high modulation efficiency), strong piezoelectricity + photoelasticity, and its wide availability. Perhaps not as clearly mentioned in the initial version of the manuscript is the appropriateness of the piezoelectric and photoelastic tensors. In particular, materials belonging to the trigonal crystal system with point group 3m (e.g. LNO) allow efficient generation of S_{yz} shear strain when the RF field is oriented along the y direction of the crystal (surface electrodes deposited on Y-cut wafer) via the d_{15} piezoelectric tensor element. This shear strain then couples strongly to an optical beam via the p_{14} photoelastic tensor element.
- Two suitable materials in the trigonal crystal system with point group 3m are: lithium niobate and lithium tantalate. These materials are optically transparent, mass manufacturable at a low-cost, and have low acoustic attenuation. When they are compared in terms of the acousto-optic figure of merit (which captures power efficiency from RF to optics), LNO is seen to be approximately an order of magnitude better than lithium tantalate.

- The manuscript is modified to address this point by adding Supplementary Information 9 (Modulator Material Choice). The reason behind using a material belonging to the trigonal crystal system with point group 3m is briefly described, the acousto-optic figure of merit is defined, and suitable materials belonging to the trigonal crystal system (lithium niobate and lithium tantalate) compared in terms of their acousto-optic figure of merit.
- New references for the Supplementary Information are also added (used in calculating the acousto-optic figures of merit for the different materials):
 1. D Pinnow. Guide lines for the selection of acoustooptic materials. IEEE Journal of Quantum Electronics, 6(4):223–238, 1970
 2. Bohdan Mytsyk, Nataliya Demyanyshyn, Anatoliy Andrushchak, and Oleh Buryy. Photoelastic properties of trigonal crystals. Crystals, 11(9):1095, 2021.
 3. Jin Yang, Jianping Long, and Lijun Yang. First-principles investigations of the physical properties of lithium niobate and lithium tantalate. Physica B: Condensed Matter, 425:12–16, 2013.

With these points addressed the manuscript could be considered further for publication in Nature Communications.

Reviewer #2 (Remarks to the Author):

The manuscript reports on a photoelastic modulator based on piezoelectrically driven bulk acoustic waves (BAWs) propagating along the Y direction of a Y-cut LiNbO₃ wafer. In the modulator, acoustic vibrations with a strong shear component and half-wavelength approximately equal to the wafer thickness are piezoelectric excited via semi-transparent contacts deposited on opposite surfaces of the wafer. The LiNbO₃ crystal is optically anisotropic in the plane perpendicular to the BAW propagation. The vibrations modulate in anti-phase the dielectric properties along two orthogonal directions on this plane at the BAW frequency, thus turning the structure into a time-dependent optical retarder with retardation controlled by the BAW amplitude. By using polarizers, the modulation of the retardation can be turned into a modulation of the transmitted intensity. The main advantages of the modulation concept are, apart from the simple design, the broad wavelength range, the high modulation efficiency, and the wide optical acceptance angle. As an example of application, the modulator is used for time-of-flight imaging with cm resolution.

The experimental work as well as the extensive modelling and simulations used for data interpretation fully support the conclusions drawn in the manuscript. Also, the authors give an illustrative application example and also compare their approach with competing ones.

- We thank the reviewer for taking their time to review our work.

The principles of operation of the proposed modulator (based on conventional acousto-optics) are, however, very well established.

- We agree with the reviewer; our work is described by the established physics of acousto-optic interactions.
- However, the device demonstrated in this work belongs to a new class of modulators with unique advantages, rather than an improvement on (or a simple extension of) existing acousto-optic modulators. Specifically, the collinear interaction of the proposed photoelastic modulator (in contrast to the transverse interaction mechanism in standard photoelastic modulators - reference 20 in the main manuscript) allows us to break the trade-off between input aperture size and the modulation frequency. Existing photoelastic modulators are severely limited in the input aperture (sub 1 mm diameter) if megahertz modulation frequencies are targeted (in high demand for a plethora of applications).
- Our unique design breaks in-plane symmetry of LN using a collinear modulation mechanism (due to the suitability of the piezoelectric and photoelastic tensors of LN - explained in the added Supplementary Information 9), allowing us to overcome the trade-off between input aperture and modulation frequency; a critical step forward, especially for ToF imaging relying on standard CMOS image sensors.
- Specifically, compared to state-of-the-art photoelastic modulators, our device operates at almost two orders of magnitude higher frequencies (and potentially up to three orders of magnitude - details in the next section), while offering a similar input aperture. Achieving this sort of advance in an established field such as acousto-optics is something that we believe has broad interest in the photonics community, as well as in optical sensing and imaging.
- Additionally, using the demonstrated photoelastic modulator for ToF imaging using standard CMOS image sensors is new. This has huge potential for ubiquitous depth sensing, since it relies on standard CMOS image sensors, which offer high performance (spatial resolution) at a low-cost.

Although the device offers advantages regarding some properties (e.g., wavelength range and acceptance angles), the operation frequencies are rather modest (few MHz, depending on crystal thickness) and cannot be easily scaled up, thus implying, e.g., in low spatial resolution for time-of-flight imaging).

- The reviewer raises an important question regarding the modulation frequency of the modulator. Indeed, the modulation frequency reported in this work is modest (~ 3.7 MHz), especially for ToF imaging applications, which usually operate in the vicinity of 20 MHz (and sometimes higher) modulation frequencies by relying on specialized image sensor pixels (e.g. photonic mixer device - reference 35 in the main manuscript).
- The work in this manuscript is a proof-of-concept, demonstrating the working principle of these new class of modulators ("longitudinal piezoelectric resonant photoelastic

modulator”), and also their enormous potential use case for converting standard CMOS image sensors (used for many applications and numbering in billions) effectively into depth sensors, and therefore adding the missing depth information to these ubiquitous sensors using a scalable and unique design. This approach has the potential of offering high performance at a low-cost, presenting an alternative to existing ToF sensors relying on specialized designs with limited spatial resolution. With our simple design, any standard image sensor can effectively become a ToF system with greater than 1 million pixels, offering for the first time higher spatial resolution even compared to specialized depth sensors (references 36 and 37 in the main manuscript). Moreover, since the proposed system relies on standard CMOS image sensors, it will reap the benefits of Moore’s law scaling and advancements in image sensors in general.

- In our later studies of the proposed modulator, we have found out that we can reach higher modulation frequencies (upto 20 MHz, needed for high performance depth sensors - see Supplementary Information 10 and 11, and Fig. S11, S13, S14) through the use of thinner LN wafers (offered commercially by optics vendors). The limiting factor on the modulation frequency is the modulation efficiency (which scales linearly with modulation frequency; holds true for any optical intensity modulator):

Line 242 in Supplementary Information - “The resonant frequency of the fundamental S_{yz} shear strain mode is inversely proportional to the wafer thickness to first order. Higher intensity modulation frequencies can be reached by using thinner LN wafers. If the wafer thickness is halved, the excited mode volume is also halved, but the average shear strain required to reach maximum intensity modulation contrast is doubled. The power in the shear strain S_{yz} is linearly proportional to the mode volume of the excited region multiplied by the average shear strain squared S_{yz}^2 . Therefore, the power required to excite the modulator for reaching maximum intensity modulation contrast is linearly proportional to the resonance frequency of the wafer to first order (assuming the acoustic quality factor remains the same for different thicknesses of the wafer). To reduce the power consumption of the modulator operating at higher frequencies, the quality factor could be increased. However, if the cut angle of the wafer is not changed, this will result in the electrical equivalent resistance to decrease at the resonance frequency, making impedance matching difficult for small electrical equivalent resistances at resonance (especially for high quality factors). To increase the quality factor while keeping the electrical equivalent resistance matched to 50Ω , a cut closer to the z axis of the wafer can be used.”

- The RF power consumption is a critical figure of merit, and should ideally not exceed several Watts to make the system compact and affordable.
- The quality factor of the demonstrated modulator in this work is approximately 1,000. However, BAW LN modulators in the gigahertz frequency regime have been reported to offer significantly higher quality factors. Quality factors of 30,000 and higher have been demonstrated, with quality factor resonance frequency products exceeding 10^{13} - references 13 and 14 in the Supplementary Information, which would improve the power

efficiency by a factor of 30. Indeed, both the efficiency and the modulation frequency can be scaled up significantly for the reported modulators by improving the design and optimizing electrode thickness and uniformity.

Line 255 in Supplementary Information:

“Quality factor greater than 30,000 even at gigahertz frequencies is achievable for LN devices, with quality factor resonance frequency product of 10^{13} and higher.”

- Below shown are COMSOL simulations for different thicknesses of Y89 LN wafer (commercially available from optics vendors). It is assumed that the quality factor is 30,000 for the three different thicknesses, which can be reached, as evidenced by references 13 and 14 in the Supplementary Information.

Fig. S11. Higher Frequency of Operation Using Thinner Wafers. **a** LN Y89 wafer with a diameter of 25.4 mm and with thickness 0.16 mm is used. Top and bottom surface electrodes are centered on the wafer, having a diameter of 12.7 mm. S_{yz} strain profile for the plane that is parallel and 0.08 mm above the bottom wafer surface is shown when 2Vpp at a frequency of 11.1106 MHz is applied to the wafer surface electrodes. **b** Simulated $|s_{11}|$ of the wafer described in **(a)** with respect to 50 Ω , with the light blue region showing the excited mode. **c** LN Y89 wafer with a diameter of 25.4 mm and with thickness 0.135 mm is used. Top and bottom surface electrodes are centered on the wafer, having a diameter of 12.7 mm. S_{yz} strain profile for the plane that is parallel and 0.0675 mm above the bottom wafer surface is shown when 2Vpp at a frequency of 13.1611 MHz is applied to the wafer surface electrodes. **d** Simulated $|s_{11}|$ of the wafer described in **(c)** with respect to 50 Ω , with the light blue region showing the excited mode. **e** LN Y89 wafer with a diameter of 25.4 mm and with thickness 0.088 mm is used. Top and bottom surface electrodes are centered on the wafer, having a diameter of 12.7 mm. S_{yz} strain profile for the plane that is parallel and 0.044 mm above the bottom wafer surface is shown when 2Vpp at a frequency of 20.1592 MHz is applied to the wafer surface electrodes. **f** Simulated $|s_{11}|$ of the wafer described in **(e)** with respect to 50 Ω , with the light blue region showing the excited mode.

- Supplementary Information 10 is added to explain how to reach higher modulation frequencies. COMSOL simulation results show that uniform strain mode, with impedance well matched to 50 Ω can be reached, while the power consumption remains at several Watts.

Line 265 in Supplementary Information:

Maximum intensity modulation is reached when $\bar{\phi}_D = \frac{4\pi L p_{14} n_o^3 \bar{S}_{yz}}{\lambda} = 1.2$, where $\lambda = 532$ nm. For the wafer of thickness $L = 0.16$ mm, $\bar{S}_{yz} \approx 5.2 \times 10^{-4}$ to reach optimum modulation. This optimum strain level is reached if $6.9 \text{ mW} \times \left(\frac{5.2 \times 10^{-4}}{3.5 \times 10^{-5}}\right)^2 \approx 1.5$ W of RF power is used to excite the surface electrodes at the resonance frequency. For the wafer of thickness $L = 0.135$ mm, $\bar{S}_{yz} \approx 6.2 \times 10^{-4}$ to reach optimum modulation. This optimum strain level is reached if $11.9 \text{ mW} \times \left(\frac{6.2 \times 10^{-4}}{4.6 \times 10^{-5}}\right)^2 \approx 2.2$ W of RF power is used to excite the surface electrodes at the resonance frequency. For the wafer of thickness $L = 0.088$ mm, $\bar{S}_{yz} \approx 9.5 \times 10^{-4}$ to reach optimum modulation. This optimum strain level is reached if $12.6 \text{ mW} \times \left(\frac{9.5 \times 10^{-4}}{4.8 \times 10^{-5}}\right)^2 \approx 4.9$ W of RF power is used to excite the surface electrodes at the resonance frequency.

[redacted]

[redacted]

[redacted]

Furthermore, the integration with other functionalities is hindered by the need of crystals with well-defined orientations.

- Specific orientations will be required for reaching different modulation frequencies to target different application spaces/needs. However, well-defined orientations of lithium niobate is a mature technology, and commercially available. For example, along with X, Y, and Z cuts, Y36, Y128 and other cut angles are commonly used for making piezoelectric devices.
1. Turutin, Andrei V., et al. "Low-frequency magnetic sensing by magnetoelectric metglas/bidomain LiNbO₃ long bars." *Journal of Physics D: Applied Physics* 51.21 (2018): 214001.
 2. Naumenko, Natalya, and Benjamin Abbott. "Optimal orientations of lithium niobate for resonator SAW filters." *IEEE Symposium on Ultrasonics, 2003*. Vol. 2. IEEE, 2003.
 3. Zhang, Naiqing, et al. "Optimized, Omnidirectional Surface Acoustic Wave Source: 152° Y-Rotated Cut of Lithium Niobate for Acoustofluidics." *IEEE Transactions on Ultrasonics, Ferroelectrics, and Frequency Control* 67.10 (2020): 2176-2186.

The results are certainly of immediate interest for the applied acousto-optic community, they lack, however, the degree of novelty and generality for a wide scientific community. For this reason, I do not recommend publication in Nat. Communication.

- The device demonstrated in this manuscript is new (rather than being a simple extension of an existing acousto-optic modulator), achieves orders of magnitude improvement over state of the art acousto-optic modulators, and achieves the highest intensity modulation efficiency among all class of intensity modulators operating in the megahertz frequency regime, while offering a simple fabrication process. Achieving this sort of improvement in an established field of research will have broad interest in the fields of optics/photonics, optical imaging and sensing.
- This manuscript is introducing the working principles of this new modulator, and its proof-of-concept demonstration. It already exceeds current acousto-optic modulators in performance by orders of magnitude, and is the most efficient free-space intensity modulator in the megahertz frequency regime. Moreover, there is still plenty of room for improvement in power efficiency and modulation frequency for the proposed modulator, as evidenced by reported quality factor values in the literature (references 13 and 14 in the Supplementary information) for lithium niobate devices, reporting quality factors greater than 30,000 even at gigahertz frequencies in LN. Additionally, recent experimental measurements of a new modulator had a quality factor of 8,000, with a resonance frequency of 7 MHz.
- The application potential for the proposed modulator, when integrated with standard CMOS image sensors, is enormous. The proposed system has the potential to convert any (and potentially all) off-the-shelf standard CMOS image sensor (numbering in the billions) effectively into a high performance ToF image sensor, adding the missing depth dimension to these ubiquitous sensors. We believe this has a very high impact and is very timely, given the emergence of autonomous systems that will greatly benefit from high performance and low-cost ToF sensors.
- Unforeseen applications may also arise through the use of the demonstrated modulators, in microscopy, and optical imaging/sensing in general.

Other suggestions/comments:

- The phase modulation along the z-direction in Fig. 1d severely restricts the useful aperture of the modulator. The authors mention that this drawback can be lifted by “depositing a static polarization manipulating metasurface” (line 266). Since the changes are dynamic, it is not clear how the metasurface can compensate for the time-dependent anti-phase ranges shown in Fig. 1d (unless, of course, the strain-dependence of the metasurface polarization is the same as in the bulk, which is very unlikely).

- The anti-phase regions (along the z-direction) can be corrected by depositing static polarization manipulating optical coatings. This can be understood by looking at Equation (2) in the main manuscript. For the red region in Fig. 1d, the intensity modulation of a laser beam propagating through this region can be expressed as follows:

$$I(t) = \frac{I_0}{2} \left(c_o^4 + c_e^4 + 2c_o^2 c_e^2 [\cos(\phi_s)(J_0(\phi_D) - 2\sin(\phi_s)\mathbf{J}_1(\phi_D)\cos(2\pi\mathbf{f}_c t) + \text{HOH})] \right)$$

➤ For the blue region, it can be expressed as follows:

$$\begin{aligned} I(t) &= \frac{I_0}{2} \left(c_o^4 + c_e^4 + 2c_o^2 c_e^2 [\cos(\phi_s)(J_0(\phi_D) - 2\sin(\phi_s)\mathbf{J}_1(-\phi_D)\cos(2\pi\mathbf{f}_c t) + \text{HOH})] \right) \\ &= \frac{I_0}{2} \left(c_o^4 + c_e^4 + 2c_o^2 c_e^2 [\cos(\phi_s)(J_0(\phi_D) + 2\sin(\phi_s)\mathbf{J}_1(\phi_D)\cos(2\pi\mathbf{f}_c t) + \text{HOH})] \right) \\ &= \frac{I_0}{2} \left(c_o^4 + c_e^4 + 2c_o^2 c_e^2 [\cos(\phi_s)(J_0(\phi_D) - 2\sin(\phi_s + \pi)\mathbf{J}_1(\phi_D)\cos(2\pi\mathbf{f}_c t) + \text{HOH})] \right) \end{aligned}$$

- The difference is a π phase shift for the laser beam propagating through the two regions, which can be corrected by the static phase term ϕ_s (ϕ_s can be used to control the sign of the modulation term, allowing the anti-phase regions to be corrected statically). The static deposited layer introduces a π phase shift to the optical beam (at a fixed wavelength of operation) for the anti-phase regions (blue).
- The main manuscript is modified to make this more clear: Line 255 - “by controlling the sign of ϕ_s ”
- Additionally, the cut angle could be changed to excite a more uniform acoustic mode. See Fig. S11 and Supplementary Information 10.

- Line 646-647: I guess that the standard deviation in Fig. 3c is probably 1 cm (and not 1m, as listed in the text).

- The standard deviation is 1 m per pixel for the ranging accuracy; the averaged ranging accuracy across all the pixels for the two different targets is approximately 1 cm. The main manuscript has been modified to make this more clear in Fig. 3 - “The distance of targets T_1 and T_2 are estimated by averaging across their corresponding pixels, respectively. The estimated distances for T_1 and T_2 are 1.07 m and 1.96 m, respectively (averaged across all the pixels corresponding to T_1 and T_2).”
- The reason for this ranging accuracy per pixel is due to the use of a relatively low power light source in the ToF experiment (~ 10 mW), along with low RF drive power for the modulator (~ 100 mW), and small optical aperture (4 mm diameter). The calculations for these have been added as a new Supplementary Information section (11), and the developed noise model agrees very well with the experimentally obtained ranging accuracy results.

Supplementary Information section 11.1:

Line 318 in Supplementary Information - “The histogram of the phase of the beat tone at 2 Hz for the enclosed region in Fig. S12c is shown in Fig. S12d. The standard deviation of the phase estimate per pixel is approximately 0.16 radians, which translates to a ranging standard deviation of $0.16 \times c/(4\pi fm) \approx 1$ m. This experimentally obtained ranging accuracy per pixel agrees well with the estimated ranging accuracy of 1.06 m using the noise model.”

- Using the developed noise model and optimized parameters for the imaging system, the attainable ranging accuracy is added as a new Supplementary Information section 11.2 and 11.3. Centimeter ranging accuracy is attainable with the following system parameters (taking maximum permissible exposure into account for eye-safety at the emission wavelength):

Line 328 in Supplementary Information - “In this section, we will demonstrate the ranging accuracy of the proposed ToF imaging system for two different cases. In the first case, the modulation frequency $f_m = 3.77$ MHz, with the following parameters: $P = 200$ mW, $f_s = 60$ Hz (ToF frame rate of 15 Hz), $N = 10^6$, $QE = 70\%$, $\lambda_o = 905$ nm, $M = 80\%$, $L = 2$ cm, $r = 10\%$, $\sigma = 6$ (corresponding to a dark current of 360 electrons/second). In the second case, the modulation frequency $f_m = 20$ MHz, with the other parameters remaining the same.”

- The ranging accuracy for the two different cases (3.77 MHz modulation frequency and 20 MHz modulation frequency) are shown below in Fig. S13.

Fig. S13. Ranging accuracy of the ToF imaging system using single frequency operation as a function of target location. It is assumed that the illuminated target in the scene is opaque and has Lambertian reflectance. **a** 3.77 MHz modulation frequency case. **b** 20 MHz modulation frequency case.

- The modulation frequency of 3.77 MHz offers approximately 10 cm ranging accuracy up to several meters range per pixel, which can still find use cases due to the high spatial resolution offered by the standard CMOS image sensor (megapixel spatial resolution). Using 20 MHz modulation frequencies results in centimeter level ranging accuracy for several meters range, offering high ranging accuracy and spatial resolution, and making it very attractive for many depth sensing applications.
- This modulation frequency can be reached with several Watts of RF power, as explained in the previous section and in Supplementary Information 10. **The ranging performance for the 20 MHz case corresponds to a depth uncertainty of 0.32%, with 1 million pixels, and a frame rate of 15 Hz.**
- This level of ranging accuracy is comparable to existing ToF sensors that rely on specialized pixels, while offering higher spatial resolution:
 1. Reference 36 in the main manuscript: Park, Byungchoul, et al. "A 64 x 64 SPAD-Based Indirect Time-of-Flight Image Sensor With 2-Tap Analog Pulse Counters." *IEEE Journal of Solid-State Circuits* (2021). - **depth uncertainty of 0.22%, with 4096 pixels, and with a frame rate of 65 Hz**
 2. Reference 37 in the main manuscript: Keel, Min-Sun, et al. "A VGA Indirect Time-of-Flight CMOS Image Sensor With 4-Tap 7 μm Global-Shutter Pixel and Fixed-Pattern Phase Noise Self-Compensation." *IEEE Journal of Solid-State Circuits* 55.4 (2019): 889-897. - **depth uncertainty of 0.37%, with 307,200 pixels, and with a frame rate of 60 Hz**
- Additionally, taking advantage of the scalability for the proposed system, multi-frequency operation can be performed through using multiple image sensors, each integrated with different modulators operating at a different frequency. This allows us to break the trade-off between unambiguous imaging range (due to phase wrapping) and the ranging accuracy. Below shown are simulations (using the same noise model that was verified experimentally), to demonstrate the ranging performance when eye-safety is taken into account.

Fig. S14. Ranging accuracy of multi-frequency operation with intensity modulation frequencies of 11 MHz, 13 MHz, and 20 MHz. **a** Ranging accuracy of multi-frequency operation with three image sensors. **b** Zoomed in version of the ranging accuracy of high performance region (up to 30 m) in (a). **c** Simulated target location likelihood using equation (S58), assuming a target is placed 26 m away from the imaging system. Reconstruction resolution is 1 cm. Estimated target location of 26.03 m agrees well with the actual target distance of 26 m. **d** Another simulated target location likelihood using equation (S58), assuming a target is placed 26 m away from the imaging system. Reconstruction resolution is 1 cm. Estimated target location of 25.81 m agrees well with the actual target distance of 26 m. **e** Simulated target location likelihood using equation (S58), assuming a target is placed 73 m away from the imaging system. Reconstruction resolution is 1 cm. Estimated target location of 60.06 m is significantly different than the actual target distance of 73 m. **f** Another simulated target location likelihood across distance using equation (S58), assuming a target is placed 73 m away from the imaging system. Reconstruction resolution is 1 cm. Estimated target location of 50.96 m is significantly different than the actual target distance of 73 m.

- The results for multi-frequency operation show that ranging accuracy up to 30 m with 10 cm ranging accuracy is possible, comparable to state of the art depth sensors, while offering significantly higher spatial resolution in a low-cost platform.

REVIEWERS' COMMENTS

Reviewer #1 (Remarks to the Author):

The authors submitted a revised version of their manuscript. In this revision they address the points of concern raised in the reports of both reviewers.

In this reviewer's opinion the issues are adequately addressed. The work is indeed a proof of concept study and the potential of the reported approach will become clearer in the future.

This reviewer has no objections against seeing this manuscript published in Nature Communications.

Reply to Reviewers

Key to the responses:

- Reviewer comments are in black
- Response to reviewer comments are in blue

REVIEWER COMMENTS

Reviewer #1 (Remarks to the Author):

The authors submitted a revised version of their manuscript. In this revision they address the points of concern raised in the reports of both reviewers. In this reviewer's opinion the issues are adequately addressed. The work is indeed a proof of concept study and the potential of the reported approach will become clearer in the future.

This reviewer has no objections against seeing this manuscript published in Nature Communications.

We thank the reviewer for taking their time to review our manuscript, assessing our manuscript revisions positively, and for supporting its publication.

We have also addressed the editorial requirements and revised the manuscript to comply with the journal requirements.